# Prevalence and factors associated with depression, anxiety, and stress among people with disabilities during COVID-19 pandemic in Bangladesh: A cross-sectional study

Nitai Roy[1]*, Md. Bony Amin[2], Mohammed A. Mamun[3,4], Bibhuti Sarker[5,6], Ekhtear Hossain[7], Md. Aktarujjaman[2]

1 Department of Biochemistry and Food Analysis, Patuakhali Science and Technology University, Patuakhali, Bangladesh, 2 Faculty of Nutrition and Food Science, Patuakhali Science and Technology University, Patuakhali, Bangladesh, 3 CHINTA Research Bangladesh, Dhaka, Bangladesh, 4 Department of Public Health and Informatics, Jahangirnagar University, Dhaka, Bangladesh, 5 Department of Economics, University of Manitoba, Winnipeg, Canada, 6 Department of Economics, Bangabandhu Sheikh Mujibur Rahman Science and Technology University, Gopalganj, Bangladesh, 7 Department of Biological Sciences and Chemistry, Southern University and A&M College, Baton Rouge, LA, United States of America

* nitai@pstu.ac.bd

**Data Availability Statement:** All relevant data are within the paper.

## Abstract

### Background

The COVID-19 pandemic has had a profound impact on the mental health of individuals across various populations. People with disabilities (PWDs) are particularly vulnerable to these effects, yet there is a lack of studies investigating the mental health of PWDs in Bangladesh. This study aims to investigate the prevalence of and factors associated with depression, anxiety, and stress among PWDs during the COVID-19 pandemic in Bangladesh.

### Methods

Data was collected through interviews with 391 PWDs between December 2020 and February 2021. Demographic information, clinical characteristics, and scores from the Depression, Anxiety, and Stress Scale (DASS-21) were obtained. Chi-square tests and logistic regression analyses were conducted to examine the relationship between psychological measures and potential risk factors.

### Results

The prevalence was found to be 65.7% for depression, 78.5% for anxiety, and 61.4% for stress, respectively. Several factors were identified as associated with these mental health issues, including gender (male), marital status (being married), low education levels, multiple impairments, comorbid medical illnesses, poor sleep quality, rural residency, hearing disability, disability onset later in life, and testing positive for COVID-19.

**Funding:** The author(s) received no specific funding for this work.

**Competing interests:** The authors have declared that no competing interests exist.

## Conclusions

The prevalence was found to be 65.7% for depression, 78.5% for anxiety, and 61.4% for stress, respectively. Several factors were identified as associated with these mental health issues, including gender (male), marital status (being married), low education levels, multiple impairments, comorbid medical illnesses, poor sleep quality, rural residency, hearing disability, disability onset later in life, and testing positive for COVID-19.

## Introduction

Disability is a significant public health concern in Bangladesh, affecting approximately 2.8% of the population, as revealed by a recent national survey conducted among 155,035 individuals in 2021 [1]. The prevalence of disability is higher among males, with 3.3% compared to 2.3% among females. The various forms of disabilities encompass a wide range of symptoms, with physical disabilities being the most prevalent (1.2%), followed by visual disabilities (0.4%), multiple disabilities (0.3%), mental illness-related disabilities (0.2%), hearing disabilities (0.2%), intellectual disabilities (0.1%), autism spectrum disorders (0.04%), speech disabilities (0.1%), deaf-blindness (0.1%), cerebral palsy (0.1%), down syndrome (0.03%), and other disabilities (0.1%) [1]. Additionally, the prevalence of at least one disability among children under the age of five is estimated to be 2.0%, with maternal factors such as functional difficulty and unhappiness in life identified as associated factors for disability in Bangladesh [2]. Notably, individuals residing in rural areas are more likely to have disabilities (2.9%) compared to those in urban areas (2.5%), leading to adverse socioeconomic outcomes and a higher risk of living below the poverty line [1].

The COVID-19 pandemic has further compounded the challenges faced by people with disabilities (PWDs). The PWDs has been identified as a high-risk group for contracting COVID-19 due to various factors such as pre-existing medical conditions, difficulties in accessing adequate sanitation and hygiene facilities, maintaining social distancing, and limited access to public health resources [1]. The World Health Organization has reported that the PWDs experience increased rates of morbidity and mortality, particularly those with underlying health conditions related to immune system function, heart disease, diabetes, or respiratory function [3–5]. Moreover, the pandemic's impact on people with disabilities extends beyond physical health risks, as they face heightened mental health vulnerabilities [6–10].

Due to these factors, it is evident that stressful situations like the COVID-19 outbreak significantly increase the risk of developing psychological vulnerabilities [11,12], and PWDs tend to experience poorer mental health compared to those without disabilities [13]. A study conducted in the United States between February and March 2021 reported similar findings, with higher prevalence rates of anxiety or depression (56.6% for PWDs vs. 28.7% for those without disabilities), new or increased substance use (38.8% vs. 17.5%), and suicidal ideation (30.6% vs. 8.3%) [8]. Similarly, studies conducted in countries like the United States [9], the United Kingdom [4], Ethiopia [6], and Canada [10] have highlighted the adverse mental health effects experienced by individuals with disabilities during the pandemic. In Bangladesh, a qualitative study has indicated that PWDs have been significantly affected by disruptions to the economy, food security, social support, and their physical and mental well-being [14]. The combination of factors such as strict social isolation measures, disrupted routines, limited access to healthcare, increased poverty, compromised education, ill health, and lack of psychological resilience further exacerbate the challenges faced by PWDs [5].

Given the detrimental impact of the COVID-19 pandemic on the mental health of people with disabilities, there is an urgent need to gather information on the mental health burdens they encounter. This information is crucial for the development of effective psychological interventions, implementation of support programs, and formulation of mental health assessment policies in Bangladesh. However, to date, limited number of epidemiological studies have been conducted in Bangladesh to ascertain the prevalence and associated factors of common mental health problems among PWDs, whereas this study aims to fill this critical knowledge gap. The findings will contribute valuable insights to inform the development of emergency response plans, recovery strategies, and integrated healthcare services to address the mental health needs of people with disabilities during and after pandemics.

## Methods

### Study design and participants

The present cross-sectional study was conducted in five conveniently selected districts in Bangladesh: Lalmonirhat, Rangpur, Nilphamari, Barishal, and Patuakhali. Within these districts, 13 *Upazilas* (local government administrative divisions) were randomly chosen, with at least one Upazila from each district included in the data collection. The study aimed to include individuals with various disabilities as the target population.

For the purpose of this study, individuals with the following disabilities were considered: those with locomotor disability (such as amputation, paralysis, or joint deformity affecting movement), visual impairment (including serious difficulty seeing even with glasses), and hearing difficulties (including those using hearing aids) [15,16]. On the other hand, people screened for leprosy and currently under treatment were recruited as leprosy [6]. Finally, people who suffer from more than one of the aforementioned disabilities were considered to have multiple impairments. The inclusion criteria consisted of being 18 years of age or older and having one or more of the specified disabilities. Certain individuals were excluded from the sample, including those under the age of 18, those who did not complete the entire survey, those who were critically ill during the study, and those who were unable to communicate and respond to the questionnaires.

### Sample size

Since the prevalence of common mental health problems in PWDs before and during the COVID-19 pandemic in Bangladesh was unknown, a conservative estimate of 50% prevalence was used for sample size calculation. With a 95% confidence interval, the estimated sample size required was 384, using the formula $n = z^2pq/d^2$. Considering a non-response rate of 10%, the total sample size aimed for was 423 individuals. Although 500 individuals were approached for interviews, 391 agreed to participate and completed the questionnaire, resulting in a response rate of 78.2%.

### Data collection procedure

A hybrid data collection approach was employed for this study. Initially, information about PWDs was obtained from the *Union Parishad* (the local government's smallest administrative division) during the collection of disability allowance. The research team then utilized this information to reach out to the PWDs and collect data. The snowball sampling method was employed to maximize the sample size from the selected areas. Trained interviewers conducted face-to-face interviews with the participants in their homes, using administered questionnaires. Participants were thoroughly briefed about the study's purpose and assured of the

confidentiality of their information. The interviews were conducted in Bangla, the native language of both the data collectors and the participants. More importantly, participants having a hearing disability used hearing aids to improve hearing and speech comprehension while conducting the survey.

## Measures

**Demographics and clinical characteristics.** Demographic information collected included age, gender, region (rural or urban), marital status, and educational level. Clinical and behavioral measures encompassed the form of disability (multiple, physical, hearing, visual, or leprosy), onset of disability, comorbid medical illnesses, and COVID-19 infection status. Socioeconomic status was classified based on income ranges (<15000 Bangladeshi Taka ≈ 177 \$, 15000–30000 Bangladeshi Taka ≈ 177–354 \$, or more than 3000 Bangladeshi Taka ≈ 177–354 \$), and sleeping hours were categorized as normal (7–9 h), less than normal (<7 h), or more than normal (>9 h) [17].

**Depression, anxiety, and stress scale.** This study used the Depression, Anxiety, and Stress Scale (DASS-21) to measure depression, anxiety, and stress. This scale consists of a 21-item questionnaire including three subscales: 7 items each for DAS with a four-point *Likert* scale ranging from 0 ("never") to 3 ("always") [18]. Sum scores are calculated by adding the scores on the items per subscale (i.e., depression, anxiety, and stress) and multiplying them by 2. The level of symptoms was categorized as follows: normal (depression 0–9, anxiety 0–7, and stress 0–14), mild (depression 10–13, anxiety 8–9, and stress 15–18), moderate (depression 14–20, anxiety 10–14, and stress 19–25), severe (depression 21–27, anxiety 15–19, and stress 26–33) and extremely severe (depression ≥28, anxiety ≥20, and stress ≥34) [19]. The cutoff point used for depression is 14, for anxiety 10, and for stress 19. The overall Cronbach's alpha of the DASS-21 was 0.86 (for depression, anxiety, and stress subscales were 0.71, 0.73, and 0.78, respectively).

**Statistical analysis.** The SPSS version 26 (SPSS 26; IBM Corp) was used to carry out all statistical analyses. Descriptive statistics (e.g., frequencies, percentages, means) were performed, to sum up, the demographic and clinical characteristics. Chi-square tests and unadjusted, and adjusted logistics regression were performed to assess the association of psychological measures to potential risk factors. The multicollinearity of variables was also checked before entering into regression analysis. All variables were entered into the multivariable analysis. Model fitness was checked using the Hosmer-Lemeshow test for depression, anxiety, and stress, which had a *p*-value of 0.625, 0.435, and 0.796, respectively, and indicated that all models were fit. Odds ratio (ORs) and 95% confidence intervals were calculated for each variable included in the regression models. The association of variables was considered statistically significant if the two-sided *p*-value was <0.05.

## Ethical approval

The study was conducted in compliance with the guidelines of the Helsinki Declaration of 1975. The study protocol was evaluated and approved by the Research Ethical Committee (REC) of the Department of Biochemistry and Food Analysis, Patuakhali Science and Technology University, Bangladesh (Approval Number: BFA: 13/11/2021:03). Participants were fully informed about the study's purpose, data confidentiality, further utilization of the collected data, and their right to withdraw from the study at any time. For participants with hearing disabilities, hearing aids were used during the survey to enhance hearing and speech comprehension. Written informed consent was obtained from all participants or their legal guardians or legally authorized representatives.

## Results

### Demographics and clinical characteristics

As shown in **Table 1**, 65.7% of the respondents were male, 44.8% were aged between 18 and 25 years (mean age = 30.73±10.84), 44.2% were single, 77.5% were from rural areas, and 29.9% had completed a higher secondary school. About 58.1% of the subjects had a physical disability, 60.1% had a comorbid illness, 63.9% reported lower-than-normal sleeping hours, and 52.9% had a disability from birth. Of them, 6.1% had tested positive for COVID-19.

### Prevalence and severity of mental health problems

The prevalence of depression, anxiety, and stress among the respondents was 65.7%, 77.5%, and 61.4%, respectively. Considering the severity of psychiatric problems, 17.6%, 15.9%, 22%, and 27.9% of the respondents reported having suffered from mild, moderate, severe, and extremely severe depression, respectively. On the other hand, 7.7%, 16.9%, 15.3%, and 46.3% reported suffering from mild, moderate, severe, and extremely severe anxiety, respectively. Finally, the prevalence of mild, moderate, severe, and extremely severe stress was 6.6%, 5.9%, 29.4%, and 26.1%, respectively (**Fig 1**).

### Associations between the studied variables and mental health problems

**Table 2** reports the associations between the studied variables and mental health concerns (depression, anxiety, and stress). With respect to gender-based depression suffering, about 75.1% of males had depression than the females, 47.8%. Similarly, a higher prevalence of depression was found in those PWDs who were single in the relationship status ($p<0.001$), had completed secondary school ($p<0.001$), and had a comorbid medical illness ($p = 0.001$). Anxiety was significantly prevalent in those PWDs who were male ($p = 0.008$) and slept less than the recommended hours ($p = 0.001$). Whereas males ($p = 0.007$), single participants ($p = 0.002$), PWDs who had completed secondary school ($p = 0.013$), suffering from leprosy ($p = 0.001$), having a comorbid illness ($p<0.001$), and sleeping less than recommended hours ($p = 0.003$) reported a higher prevalence of stress (**Table 2**).

 **Table 3** reports the risk factors for mental disorders from a multivariable logistic regression model. Being male (AOR = 2.64; 95% CI = 1.49–4.67), completing secondary school (AOR = 4.79; 95% CI = 1.71–13.44), completing higher secondary school (AOR = 4.19; 95% CI = 1.52–11.58), having a comorbid medical illness (AOR = 2.34; 95% CI = 1.41–3.89) emerged as risk factors of depression, whereas living in a rural area (AOR = 0.46; 95% CI = 0.25–0.87), being divorced/widowed/separated (AOR = 0.30; 95% CI = 0.12–0.79], suffering from leprosy (AOR = 0.28; 95% CI = 0.08–0.90), and sleep more than normal (AOR = 0.33; 95% CI = 0.12–0.89) were negatively associated with depression. Regarding anxiety risk, living in a rural area (AOR = 2.06; 95% CI = 1.09–3.91), suffering from a hearing disability (AOR = 6.83; 95% CI = 2.07–22.56), sleeping less than normal (AOR = 3.14; 95% CI = 1.65–6.00), and tested positive for COVID-19 (AOR = 5.63; 95% CI = 1.00–31.58) emerged as independent risk factors. Moreover, individuals disabled from early childhood (AOR = 0.43; 95% CI = 0.19–0.97) were negatively associated with anxiety. The PWDs completed secondary school (AOR = 5.58; 95% CI = 1.97–15.81), completed higher secondary school (AOR = 3.40; 95% CI = 1.23–9.37), had a comorbid medical illness (AOR = 2.78; 95% CI = 1.72–4.51), and sleep less than normal (AOR = 2.31; 95% CI = 1.30–4.10) emerged as risk factors of stress, whereas being divorced/widowed/separated (AOR = 0.29; 95% CI = 0.11–0.73], and suffering from hearing disability (AOR = 0.32; 95% CI = 0.12–0.87] were negatively associated with stress.

**Table 1. Demographic and clinical characteristics.**

| Variables | Categories | Total | % |
|---|---|---|---|
| **Age group (years)** | | | |
| | 18 to 25 | 175 | 44.8 |
| | 26 to 35 | 101 | 25.8 |
| | Above 35 | 115 | 29.4 |
| **Gender** | | | |
| | Male | 257 | 65.7 |
| | Female | 134 | 34.3 |
| **Region** | | | |
| | Rural | 303 | 77.5 |
| | Urban | 88 | 22.5 |
| **Marital Status** | | | |
| | Single | 173 | 44.2 |
| | Divorced/widowed/separated | 47 | 12.0 |
| | Married | 171 | 43.7 |
| **Monthly income (BDT)** | | | |
| | Below 15000 | 223 | 57.0 |
| | 15–30000 | 126 | 32.2 |
| | Above 30000 | 42 | 10.7 |
| **Education level** | | | |
| | No formal education | 78 | 19.9 |
| | Primary school | 56 | 14.3 |
| | Secondary school | 83 | 21.2 |
| | Higher Secondary school | 117 | 29.9 |
| | Honors | 30 | 7.7 |
| | Masters or above | 27 | 6.9 |
| **Form of disability** | | | |
| | Physical disability | 227 | 58.1 |
| | Hearing disability | 62 | 15.9 |
| | Visual disability | 39 | 10.0 |
| | Leprosy | 27 | 6.9 |
| | Multiple impairments | 36 | 9.2 |
| **Onset of disability** | | | |
| | From birth | 207 | 52.9 |
| | Early childhood | 102 | 26.1 |
| | Later in life | 82 | 21.0 |
| **Comorbid medical illness** | | | |
| | Yes | 235 | 60.1 |
| | No | 156 | 39.9 |
| **Sleep status (hours)** | | | |
| | Below normal (<7) | 250 | 63.9 |
| | Above normal (>9) | 29 | 7.4 |
| | Normal (7–9) | 112 | 28.6 |
| **Positive with COVID-19** | | | |
| | Yes | 24 | 6.1 |
| | No | 367 | 93.9 |

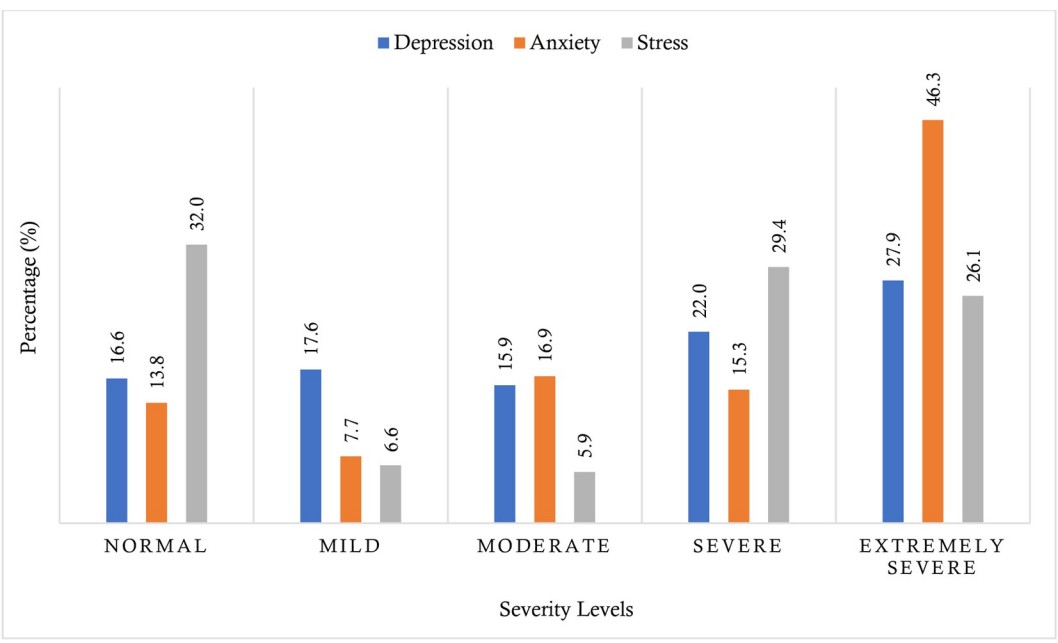

**Fig 1. The severity of depression, anxiety, and stress, among the Bangladeshi PWDs.**

## Discussion

The present study on the mental health burden of people with disabilities (PWDs) during the COVID-19 pandemic in Bangladesh contributes to the limited research conducted in this country. Globally, numerous studies have examined the mental health of PWDs during the pandemic, but Bangladesh has been relatively underrepresented in the literature [6,8–10,13,20]. Therefore, this study fills an important knowledge gap by shedding light on the psychological challenges faced by PWDs in Bangladesh. As per the findings of this study, the prevalence of depression, anxiety, and stress among PWDs was found to be 65.7%, 78.5%, and 61.4%, respectively during the COVID-19 pandemic in Bangladesh. PWDs with hearing impairment, visual impairment, and leprosy were more severely affected, as were those with comorbid illnesses, insufficient sleep, COVID-19 infection, and higher mental health burden. Socio-demographic factors such as age, gender, region, marital status, and education were identified as predictors of mental health issues. These findings highlight the need to consider specific risk factors when planning mental health programs and interventions for PWDs.

Comparing the findings of this study with previous research, it is noteworthy that the prevalence rates of depression and anxiety among PWDs in Ethiopia were lower (46.2% and 48.1% respectively) [6]. Similarly, in the United States, adults with cognitive and visual disabilities exhibited a slightly lower rate of depression (76.4%) compared to the current study [9]. Another study conducted in the US reported lower rates of major depressive disorder (61.0%) and generalized anxiety disorder (50.0%) among PWDs [21]. In contrast, a recent systematic review of various Bangladeshi population during the COVID-19 pandemic found lower pooled prevalence rates of depression, anxiety, and stress (47%, 47%, and 44% respectively [22]. The higher mental health burdens observed in the present study among PWDs in Bangladesh could be attributed to specific stressors identified in this population, such as limited access to healthcare services, emotional abuse, difficulties in managing chronic conditions, and lack of sufficient support [9]. These factors may contribute to the heightened psychological distress experienced by PWDs during the pandemic. Overall, this study underscores the importance of

**Table 2. Association between study variables and depression, anxiety, and stress among people with disabilities during the COVID-19 pandemic.**

| Variables | | Depression | | | Anxiety | | | Stress | | |
|---|---|---|---|---|---|---|---|---|---|---|
| | | Yes (%) | $\chi^2$ | p-value | Yes (%) | $\chi^2$ | p-value | Yes (%) | $\chi^2$ | p-value |
| **Age group (years)** | | | | | | | | | | |
| | 18 to 25 | 124 (70.9) | 3.82 | 0.148 | 144 (82.3) | 5.67 | 0.059 | 111 (63.4) | 0.78 | 0.676 |
| | 26 to 35 | 61 (60.4) | | | 71 (70.3) | | | 62 (61.4) | | |
| | Above 35 | 72 (62.6) | | | 92 (80.0) | | | 67 (58.3) | | |
| **Gender** | | | | | | | | | | |
| | Male | 193 (75.1) | 29.22 | **<0.001** | 212 (82.5) | 7.02 | **0.008** | 170 (66.1) | 7.19 | **0.007** |
| | Female | 64 (47.8) | | | 95 (70.9) | | | 70 (52.2) | | |
| **Region** | | | | | | | | | | |
| | Rural | 192 (63.4) | 3.34 | 0.068 | 244 (80.5) | 3.23 | 0.072 | 188 (62.0) | 0.25 | 0.616 |
| | Urban | 65 (73.9) | | | 63 (71.6) | | | 52 (59.1) | | |
| **Marital status** | | | | | | | | | | |
| | Single | 127 (73.4) | 20.24 | **<0.001** | 145 (83.8) | 5.17 | 0.076 | 119 (68.8) | 12.93 | **0.002** |
| | Divorced/widowed/separated | 18 (38.3) | | | 35 (74.5) | | | 19 (40.4) | | |
| | Married | 112 (65.5) | | | 127 (74.3) | | | 102 (59.6) | | |
| **Monthly income (BDT)** | | | | | | | | | | |
| | Below 15000 | 142 (63.7) | 1.05 | 0.592 | 176 (78.9) | 2.70 | 0.260 | 138 (61.9) | 0.36 | 0.837 |
| | 15–30000 | 87 (69.0) | | | 102 (81.0) | | | 78 (61.9) | | |
| | Above 30000 | 28 (66.7) | | | 29 (69.0) | | | 24 (57.1) | | |
| **Education level** | | | | | | | | | | |
| | No formal education | 47 (60.3) | 27.84 | **<0.001** | 59 (75.6) | 9.28 | 0.098 | 44 (56.4) | 14.50 | **0.013** |
| | Primary school | 26 (46.4) | | | 40 (71.4) | | | 33 (58.9) | | |
| | Secondary school | 66 (79.5) | | | 73 (88.0) | | | 59 (71.1) | | |
| | Higher Secondary school | 88 (75.2) | | | 95 (81.2) | | | 79 (67.5) | | |
| | Honors | 18 (60.0) | | | 21 (70.0) | | | 15 (50.0) | | |
| | Masters or above | 12 (44.4) | | | 19 (70.4) | | | 10 (37.0) | | |
| **Form of disability** | | | | | | | | | | |
| | Physical disability | 155 (68.3) | 9.26 | 0.055 | 174 (76.7) | 7.71 | 0.103 | 143 (63.0) | 17.68 | **0.001** |
| | Hearing disability | 44 (71.0) | | | 55 (88.7) | | | 25 (40.3) | | |
| | Visual disability | 24 (61.5) | | | 32 (82.1) | | | 29 (74.4) | | |
| | Leprosy | 11 (40.7) | | | 22 (81.5) | | | 21 (77.8) | | |
| | Multiple impairments | 23 (63.9) | | | 24 (66.7) | | | 22 (61.1) | | |
| **Onset of disability** | | | | | | | | | | |
| | From birth | 135 (65.2) | 1.34 | 0.512 | 167 (80.7) | 5.32 | 0.070 | 127 (61.4) | 1.05 | 0.592 |
| | Early childhood | 64 (62.7) | | | 72 (70.6) | | | 66 (64.7) | | |
| | Later in life | 58 (70.7) | | | 68 (82.9) | | | 47 (57.3) | | |
| **Comorbid medical illness** | | | | | | | | | | |
| | Yes | 170 (72.3) | 11.43 | **0.001** | 191 (81.3) | 2.66 | 0.103 | 167 (71.1) | 23.30 | **<0.001** |
| | No | 87 (55.8) | | | 116 (74.4) | | | 73 (46.8) | | |
| **Sleep status (hours)** | | | | | | | | | | |
| | Below normal (<7) | 170 (68.0) | 4.51 | 0.105 | 210 (84.0) | 14.75 | **0.001** | 169 (67.6) | 11.78 | **0.003** |
| | Above normal (>9) | 14 (48.3) | | | 23 (79.3) | | | 13 (44.8) | | |
| | Normal (7–9) | 73 (65.2) | | | 74 (66.1) | | | 58 (51.8) | | |
| **Positive with COVID-19** | | | | | | | | | | |
| | Yes | 13 (54.2) | 1.52 | 0.218 | 22 (91.7) | 2.62 | 0.105 | 19 (79.2) | 3.41 | 0.065 |
| | No | 244 (66.5) | | | 285 (77.7) | | | 221 (60.2) | | |

p-values calculated using Chi-square analysis.

**Table 3. Factors associated with depression, anxiety, and stress among people with disabilities during the COVID-19 pandemic.**

| Variables | Categories | Depression | | Anxiety | | Stress | |
|---|---|---|---|---|---|---|---|
| | | *p*-value | AOR (LL-UL) | *p*-value | AOR (LL-UL) | *p*-value | AOR (LL-UL) |
| **Age group (years)** | | | | | | | |
| | 18 to 25 | 0.495 | 0.75 [0.34–1.69] | 0.451 | 0.70 [0.28–1.77] | 0.283 | 0.66 [0.31–1.41] |
| | 26 to 35 | 0.754 | 0.90 [0.45–1.78] | 0.120 | 0.56 [0.27–1.16] | 0.977 | 1.01 [0.53–1.94] |
| | Above 35 | | Reference | | Reference | | Reference |
| **Gender** | | | | | | | |
| | Male | **0.001** | 2.64 [1.49–4.67] | 0.067 | 1.79 [0.96–3.36] | 0.728 | 1.10 [0.63–1.94] |
| | Female | | Reference | | Reference | | Reference |
| **Region** | | | | | | | |
| | Rural | **0.016** | 0.46 [0.25–0.87] | **0.027** | 2.06 [1.09–3.91] | 0.820 | 1.07 [0.61–1.89] |
| | Urban | | Reference | | Reference | | Reference |
| **Marital Status** | | | | | | | |
| | Single | 0.420 | 1.37 [0.64–2.91] | 0.293 | 1.58 [0.67–3.74] | 0.182 | 1.64 [0.79–3.37] |
| | Divorced/widowed/separated | **0.015** | 0.30 [0.12–0.79] | 0.730 | 0.83 [0.29–2.39] | **0.009** | 0.29 [0.11–0.73] |
| | Married | | Reference | | Reference | | Reference |
| **Monthly income (BDT)** | | | | | | | |
| | Below 15000 | 0.449 | 1.38 [0.60–3.16] | 0.132 | 1.89 [0.83–4.35] | 0.763 | 1.13 [0.52–2.43] |
| | 15–30000 | 0.467 | 1.38 [0.58–3.28] | 0.063 | 2.32 [0.96–5.61] | 0.894 | 0.95 [0.42–2.12] |
| | Above 30000 | | Reference | | Reference | | Reference |
| **Education level** | | | | | | | |
| | No formal education | 0.253 | 1.82 [0.65–5.07] | 0.910 | 0.94 [0.31–2.86] | 0.417 | 1.54 [0.54–4.36] |
| | Primary school | 0.947 | 0.96 [0.33–2.79] | 0.658 | 0.77 [0.25–2.42] | 0.189 | 2.07 [0.70–6.16] |
| | Secondary school | **0.003** | 4.79 [1.71–13.44] | 0.088 | 2.75 [0.86–8.78] | **0.001** | 5.58 [1.97–15.81] |
| | Higher Secondary school | **0.006** | 4.19 [1.52–11.58] | 0.863 | 1.10 [0.37–3.26] | **0.018** | 3.40 [1.23–9.37] |
| | Honors | 0.299 | 1.89 [0.57–6.32] | 0.487 | 0.64 [0.18–2.28] | 0.456 | 1.57 [0.48–5.17] |
| | Masters or above | | Reference | | Reference | | Reference |
| **Form of disability** | | | | | | | |
| | Physical disability | 0.882 | 0.94 [0.39–2.26] | 0.061 | 2.39 [0.96–5.96] | 0.891 | 1.06 [0.45–2.48] |
| | Hearing disability | 0.864 | 1.10 [0.38–3.14] | **0.002** | 6.83 [2.07–22.56] | **0.025** | 0.32 [0.12–0.87] |
| | Visual disability | 0.489 | 0.66 [0.20–2.14] | 0.073 | 3.24 [0.90–11.72] | 0.223 | 2.08 [0.64–6.76] |
| | Leprosy | **0.032** | 0.28 [0.08–0.90] | 0.128 | 2.90 [0.73–11.48] | 0.470 | 1.58 [0.45–5.53] |
| | Multiple impairments | | Reference | | Reference | | Reference |
| **Onset of disability** | | | | | | | |
| | From birth | 0.154 | 0.61 [0.31–1.21] | 0.979 | 1.01 [0.47–2.18] | 0.471 | 1.26 [0.67–2.35] |
| | Early childhood | 0.137 | 0.56 [0.27–1.20] | **0.041** | 0.43 [0.19–0.97] | 0.603 | 1.21 [0.60–2.44] |
| | Later in life | | Reference | | Reference | | Reference |
| **Comorbid medical illness** | | | | | | | |
| | Yes | **0.001** | 2.34 [1.41–3.89] | 0.304 | 1.34 [0.77–2.32] | **<0.001** | 2.78 [1.72–4.51] |
| | No | | Reference | | Reference | | Reference |
| **Sleep status** | | | | | | | |
| | Below normal (<7) | 0.153 | 1.57 [0.85–2.90] | **0.001** | 3.14 [1.65–6.00] | **0.004** | 2.31 [1.30–4.10] |
| | Above normal (>9) | **0.029** | 0.33 [0.12–0.89] | 0.205 | 2.00 [0.69–5.82] | 0.202 | 0.54 [0.21–1.40] |
| | Normal (7–9) hours | | Reference | | Reference | | Reference |
| **Positive with COVID-19** | | | | | | | |
| | Yes | 0.109 | 0.42 [0.15–1.21] | **0.049** | 5.63 [1.00–31.58] | 0.434 | 1.60 [0.49–5.19] |
| | No | | Reference | | Reference | | Reference |

AOR = Adjusted odds ratio, LL = Lower limit, UL = Upper limit.

understanding the unique challenges faced by PWDs during the COVID-19 pandemic and highlights the need for tailored mental health support and interventions. By considering the specific stressors and risk factors identified in this study, policymakers and healthcare authorities can develop targeted programs and allocate resources effectively.

The findings of this study reveal an interesting gender difference in the prevalence of mental health disorders among PWDs, with male individuals being more susceptible. While this distribution aligns with previous study [23], it emphasizes the need for intervention programs to prioritize the mental health needs of male PWDs. By recognizing and addressing this gender-specific vulnerability, tailored support and interventions can be implemented to better meet the unique challenges faced by male PWDs. In contrast to the recent studies [24,25], this study uncovered a noteworthy association between residing in urban areas and experiencing higher rates of depression among PWDs. This finding diverges from the commonly observed trend where urban areas offer better access to resources and support services. The possible explanation for this discrepancy could be linked to the flow of information during the pandemic. Urban areas tend to have higher exposure to negative news and heightened anxiety levels due to the densely populated nature and the constant need for individuals to confirm their potential exposure to COVID-19. In contrast, rural areas provided a more conducive environment for physical activities and stronger social bonding, potentially leading to lower rates of anxiety among rural PWDs.

This study found that married PWDs had a higher level of depression compared to those who were divorced, widowed, or separated, which is consistent with findings from other studies [10]. However, it has also been observed in another study that being married can reduce the risk of developing depression among PWDs [6]. This could be because having a partner facilitates socializing, especially during the pandemic when there are restrictions on movement and limited opportunities for normal social activities. Additionally, individuals who have experienced the loss of a partner may have developed resilience to cope with the challenges arising from the pandemic.

Furthermore, education plays a significant role, particularly during critical periods such as the COVID-19 pandemic. Study has shown that individuals with no formal education exhibit lower levels of COVID-19 preventive behaviors [26]. The reduced concern for preventing infection among these individuals can lead to a sense of vulnerability to the virus, potentially contributing to mental instability. This study supports previous findings [6,20] by highlighting that PWDs with lower educational attainment are at a higher risk of experiencing mental health issues. On the other hand, higher educational attainment equips PWDs with knowledge about preventive measures, enabling them to protect themselves and potentially reduce the development of mental health symptoms. Education fosters a sense of mastery and self-esteem, which can positively impact mental well-being in PWDs.

The presence of comorbid medical conditions was found to significantly contribute to depression and stress among the sample of PWDs in this study. This finding aligns with a US study that revealed PWDs faced challenges in managing their chronic conditions compared to those without disabilities [9], which subsequently increased their risk of experiencing mental health problems [9,10]. The heightened vulnerability of individuals with comorbid medical illnesses to COVID-19 may explain this association, as it can lead to heightened psychological vulnerability. Furthermore, a study conducted in the UK during the COVID-19 pandemic found the association between anxiety and stress, and poor sleep status among individuals with visual disabilities [27]. Similarly, in this study, PWDs who reported sleeping less than their normal pattern exhibited higher levels of anxiety and stress compared to those who maintained normal sleep patterns. Sleep disturbances can significantly impact mental well-being,

and this association underscores the importance of addressing sleep issues in PWDs to mitigate their psychological distress during the pandemic.

In this study, the relationship between the type of disabilities and mental health problems did not exhibit significant differences. However, findings from an Ethiopian study indicated that PWDs with hearing disabilities and leprosy experienced more severe depression and insomnia symptoms [6]. Similarly, a UK study concluded that individuals with hearing difficulties reported increased levels of self-reported anxiety (35.4% reported increased anxiety) and depression (30% reported worsening depression) during the COVID-19 pandemic compared to pre-pandemic data [20]. Despite the existing evidence, the current study did not find the type of disabilities or the onset of disabilities to be significant predictors of mental health problems. However, it is important to note that further studies should specifically focus on examining the contributing roles of these factors in mental health outcomes among PWDs. By delving deeper into the relationship between the type of disabilities, onset of disabilities, and mental health, we can gain a better understanding of how these factors interact and influence mental well-being in this population.

This study has several limitations that should be acknowledged. Firstly, due to its cross-sectional design, causal relationships cannot be established. The data collected at a single time point does not allow for determining the directionality of the relationships observed. Additionally, the convenience and snowball sampling methods employed in this study may introduce selection bias and limit the generalizability of the findings to the broader population of people with disabilities. While the mental health conditions were assessed using validated tools and structured interviews conducted by trained interviewers, it is important to note that these measures may not be as comprehensive or accurate as those performed by experienced clinicians. The use of additional surveys with larger sample sizes would enhance the reliability and validity of the comparisons made across individuals with different types of disabilities. Furthermore, collecting longitudinal data would provide a more comprehensive understanding of how perceptions and mental health outcomes change over time in response to the evolving circumstances of the COVID-19 pandemic, as well as considering other factors such as demographics and clinical characteristics. It is also worth mentioning that the absence of pre-pandemic data in this study makes it difficult to ascertain whether the observed mental health problems were directly caused by the COVID-19 pandemic or if they existed prior to the outbreak. Longitudinal studies with pre-pandemic baseline measurements would be valuable in addressing this issue. Lastly, the study participants were limited to PWDs from five districts in Bangladesh, which may restrict the generalizability of the findings to other regions or countries. Further research involving a more diverse and representative sample would provide a broader perspective on the mental health challenges faced by PWDs during the pandemic.

This study holds significant importance as it is the first of its kind in Bangladesh to explore the psychological burden and associated factors among PWDs during a pandemic. Given the scarcity of available evidence in the country, this study provides crucial and supportive information about a marginalized community during an infectious outbreak. The findings of this study are expected to stimulate the development of appropriate interventions and strategies to assist and manage psychological difficulties, ultimately improving the mental health of people with disabilities. It is crucial for Bangladesh's health authorities to recognize that PWDs are a high-risk group vulnerable to psychological distress during the pandemic. By acknowledging this, they can implement targeted measures to address the specific needs of this population, taking into account various social distancing measures. The insights gained from this research can be utilized to inform policymaking, resource allocation, and the establishment of accessible services. Moreover, the study findings can be instrumental in advocating for better mental health services and promoting the rights and well-being of PWDs in Bangladesh. Overall, this

study's significance lies in filling the knowledge gap regarding the psychological well-being of PWDs during a pandemic in Bangladesh. It provides a foundation for future research and underscores the importance of addressing the mental health needs of this vulnerable population through evidence-based interventions and policies.

## Conclusions

In conclusion, this study sheds light on the prevalence of common mental health problems among people with disabilities (PWDs) during the COVID-19 pandemic. The findings highlight the significant psychological burden faced by many PWDs, potentially exacerbated by pandemic-related stressors. The results underscore the urgent need for targeted interventions and support systems to address the mental health needs of PWDs. It is crucial for health authorities and policymakers to prioritize the well-being of PWDs, ensuring that their basic psychological needs are met and effective strategies are developed to improve their mental health outcomes. Additionally, the healthcare system must recognize and overcome the existing limitations in providing comprehensive mental health support to PWDs. Accessible and inclusive mental health services should be made available, and PWDs should be given equitable opportunities to receive the necessary care and support. Furthermore, it is imperative for the government and relevant stakeholders to advocate for the rights and well-being of PWDs, both during the pandemic and beyond. This includes ensuring that PWDs have equal access to healthcare services and support systems, as well as addressing the broader societal factors that contribute to their mental health challenges.

## Acknowledgments

We greatly thank all the participants and data collectors of the study.

## Author Contributions

**Conceptualization:** Nitai Roy.

**Data curation:** Nitai Roy, Md. Bony Amin, Md. Aktarujjaman.

**Formal analysis:** Nitai Roy, Md. Bony Amin.

**Investigation:** Nitai Roy, Md. Bony Amin.

**Methodology:** Nitai Roy, Md. Bony Amin, Md. Aktarujjaman.

**Software:** Nitai Roy, Md. Bony Amin.

**Supervision:** Nitai Roy.

**Validation:** Nitai Roy, Md. Bony Amin, Mohammed A. Mamun, Bibhuti Sarker, Ekhtear Hossain.

**Visualization:** Nitai Roy, Md. Bony Amin, Mohammed A. Mamun, Bibhuti Sarker, Ekhtear Hossain.

**Writing – original draft:** Nitai Roy, Md. Bony Amin.

**Writing – review & editing:** Nitai Roy, Md. Bony Amin, Mohammed A. Mamun, Bibhuti Sarker, Ekhtear Hossain, Md. Aktarujjaman.

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
