## [Decision Letter · Decision Letter 0]

8 May 2023

PONE-D-23-05590Prevalence and factors associated with depression, anxiety, and stress among people with disabilities during COVID-19 pandemic in Bangladesh: a cross-sectional studyPLOS ONE

Dear Dr. Roy,

Thank you for submitting your manuscript to PLOS ONE. After careful consideration, we feel that it has merit but does not fully meet PLOS ONE’s publication criteria as it currently stands. Therefore, we invite you to submit a revised version of the manuscript that addresses the points raised during the review process.

We look forward to receiving your revised manuscript.

Kind regards,

Shafiun Nahin Shimul

Academic Editor

PLOS ONE

Reviewers' comments:

Reviewer's Responses to Questions

**Comments to the Author**

1. Is the manuscript technically sound, and do the data support the conclusions?

Reviewer #1: Yes

Reviewer #2: Yes

2. Has the statistical analysis been performed appropriately and rigorously? 

Reviewer #1: Yes

Reviewer #2: Yes

3. Have the authors made all data underlying the findings in their manuscript fully available?

Reviewer #1: Yes

Reviewer #2: Yes

4. Is the manuscript presented in an intelligible fashion and written in standard English?

Reviewer #1: Yes

Reviewer #2: Yes

5. Review Comments to the Author

Reviewer #1: I am glad to have had the opportunity to review this pertinent and interesting paper. The paper addresses a relevant issue, both socially and scientifically.

Please find my comments below:

1. Introduction is well written with enough supporting citations. However, the last paragraph of the introduction could be more specific about the policy implications of the study.

2.The method is adequate. Sample size selection and description of the data collection procedure seem perfect to me. Description of the variables are clarified ( there are some typos in this section which should be corrected carefully). However, in this section the referencing style is different from the other sections. This issue has to be taken care of.

3.The statistical strategy used is relevant and allows to get the objectives of the study.

4. Regarding results, in general, the section is clear.

5. The discussion and conclusion parts are well described.

Reviewer #2: General comments

The manuscript introduces the mental health consequences of the COVID 19 pandemic, and I believe it would be of interest to the readership of Plos One. However, the manuscript will need some major revisions.

The findings should be interpreted with caution because of the approach (convenience and snowball sampling). This should be acknowledged in the limitations of the study. Anxiety and depression usually have similar risk factors but the findings from your regression models were the reverse. Did you run an interaction term to check if there is evidence of presence of interaction between some of your independent variables (e.g region, income, and education)?

The discussion section should be restructured. In many places, you diligently provide detailed information on findings from previous studies. This shows your attention to detail, but it is usually unnecessary and detracts the reader. It's better to summarize the key findings across studies to show differences and similarities, i.e., synthesize the results rather than just giving a laundry list of studies. Make sure each paragraph is focused on a specific topic, and stay on it, without introducing other things.

Please take note of the following:

Page 1 line 27, write DASS in full the first time it is used in the manuscript.

Page 3 line 42, mention the year and the population covered in the recent national survey.

Page 3 line 43, the figures mentioned are prevalence and not “Prevalence rates.”

Some of the percentages/proportions have one decimal place while others have two. Be consistent in the use of one or two decimal points throughout the document.

What age cohort was covered int the MICS survey conducted in 2019 (line 49, first paragraph)? How different is it from the national survey?

Page 3 lines 52-53, there is not much difference between rural and urban areas based on the figures reported (2.9% vs 2.5 %)? How precise are those estimates, what are the confidence intervals and/or p-value?

Lines 54-62- The sentences are very long with several grammatical errors. The whole paragraph should be revised.

Based on the information provided in lines 63-78, the association between the COVID 19 pandemic and mental health is well established. What does your study add? How different is it to other studies? What is the significance of your study in the context of Bangladesh?

Lines 180-199, what is reported in Table 2 and Figure 1 are not prevalence rates, please revise the text accordingly.

Line 263-281, the discussions on marital status are contradictory.

There are some grammatical, typo and formatting errors.

6. PLOS authors have the option to publish the peer review history of their article (what does this mean?). If published, this will include your full peer review and any attached files.

Reviewer #1: No

Reviewer #2: No

---

## [Author Response · Author response to Decision Letter 0]

16 Jun 2023

Manuscript ID: PONE-D-23-05590

Manuscript Title: Prevalence and factors associated with depression, anxiety, and stress among people with disabilities during COVID-19 pandemic in Bangladesh: a cross-sectional study

Journal: PLOS ONE

Comments to the Author

1. Is the manuscript technically sound, and do the data support the conclusions?

Reviewer #1: Yes

Reviewer #2: Yes

Response: Thank you so much for affirming the technical quality of our manuscript. 

2. Has the statistical analysis been performed appropriately and rigorously?

Reviewer #1: Yes

Reviewer #2: Yes

Response: Thank you so much for clarifying that our manuscript's statistical analysis was carried out correctly and methodically.

3. Have the authors made all data underlying the findings in their manuscript fully available?

Reviewer #1: Yes

Reviewer #2: Yes

Response: Thank you so much for verifying that we provided the data availability statement as regards our manuscript.

4. Is the manuscript presented in an intelligible fashion and written in standard English?

Reviewer #1: Yes

Reviewer #2: Yes

Response: Thank you so much for the kind remarks about the English standard and manuscript presentation.

Response to the Reviewer 1’s Comments

Comment 1: I am glad to have had the opportunity to review this pertinent and interesting paper. The paper addresses a relevant issue, both socially and scientifically.

Response 1: We appreciate your positive feedback on our manuscript. Thank you for your kind words.

Comment 2: Introduction is well written with enough supporting citations. However, the last paragraph of the introduction could be more specific about the policy implications of the study.

Response 2: Thank you for identifying this issue. We have now revised the last paragraph of the introduction to provide more specific information about the policy implications of the study (Line 94-96 in the manuscript with track changes version).

Comment 3: The method is adequate. Sample size selection and description of the data collection procedure seem perfect to me. Description of the variables are clarified (there are some typos in this section which should be corrected carefully). However, in this section the referencing style is different from the other sections. This issue has to be taken care of.

Response 3: Thank you for your positive feedback on our methods. We have carefully addressed and corrected all the typos in the description of variables. Additionally, we have ensured consistency in the referencing style throughout the manuscript (Line 156, 162, and 419-424 in the manuscript with track changes version).

. 

Comment 4: The statistical strategy used is relevant and allows to get the objectives of the study.

Response 4: We appreciate your kind compliments on our statistical strategy. Thank you for recognizing its relevance in achieving the objectives of the study. 

Comment 5: Regarding results, in general, the section is clear.

Response 5: We are grateful for your positive feedback on our results. Thank you for acknowledging the clarity of the section. 

Comment 6: The discussion and conclusion parts are well described.

Response 6: Thank you for your positive feedback on our discussion and conclusion. We appreciate your kind words.

Response to the Reviewer 2’s Comments

Comment 1: The manuscript introduces the mental health consequences of the COVID 19 pandemic, and I believe it would be of interest to the readership of Plos One. However, the manuscript will need some major revisions.

Response 1: We sincerely appreciate the reviewer's valuable suggestions and comments, which have helped us in revising the manuscript accordingly. 

Comment 2: The findings should be interpreted with caution because of the approach (convenience and snowball sampling). This should be acknowledged in the limitations of the study.

Response 2: Thank you for your suggestion. In the limitations section of our publication, we have duly acknowledged the need for caution in interpreting the findings due to the convenience and snowball sampling approach (Line 329-331 in the manuscript with track changes version).

. 

Comment 3: Anxiety and depression usually have similar risk factors but the findings from your regression models were the reverse. Did you run an interaction term to check if there is evidence of presence of interaction between some of your independent variables (e.g region, income, and education)?

Response 3: Thank you so much for your comment. Yes, we did run an interaction term to check for the presence of interaction between some of our independent variables (e.g., region*education, income*education, region*income, age*gender, and type of disability*illness). However, the interaction was not statistically significant, which is why interaction terms were not included in the regression model.

Comment 4: The discussion section should be restructured. In many places, you diligently provide detailed information on findings from previous studies. This shows your attention to detail, but it is usually unnecessary and detracts the reader. It's better to summarize the key findings across studies to show differences and similarities, i.e., synthesize the results rather than just giving a laundry list of studies. Make sure each paragraph is focused on a specific topic, and stay on it, without introducing other things.

Response 4: We greatly appreciate your insightful comments. As suggested, we have restructured our discussion section by dividing the 3rd, 4th, and 5th paragraphs to ensure each paragraph focuses on a specific topic without introducing unrelated details. Additionally, we have summarized the key findings across studies in order to synthesize the results effectively rather than reporting details of the previous studies (Line 252-272, 273-288, 289-308, and 309-318 in the manuscript with track changes version).

Comment 5: Page 1 line 27, write DASS in full the first time it is used in the manuscript.

Response 5: We have addressed this issue in the revised version of the manuscript by writing DASS in full the first time it is mentioned (Line 27 in the manuscript with track changes version).

.

Comment 6: Page 3 line 42, mention the year and the population covered in the recent national survey.

Response 6: We have made the necessary revision in the manuscript, specifying the year and the population covered in the recent national survey as suggested. Thanks. (Line 43, 44 in the manuscript with track changes version).

Comment 7: Page 3 line 43, the figures mentioned are prevalence and not “Prevalence rates.”

Response 7: This correction has been made in the revised version of the manuscript. Thank you. (Line 45 in the manuscript with track changes version).

Comment 8: Some of the percentages/proportions have one decimal place while others have two. Be consistent in the use of one or two decimal points throughout the document.

Response 8: We have addressed this inconsistency in the revised version of the manuscript. However, please note that for certain conditions like autism spectrum disorder and Down syndrome, the percentages have not been adjusted, as doing so would result in a percentage of zero. (Line 46-50 in the manuscript with track changes version).

Comment 9: What age cohort was covered int the MICS survey conducted in 2019 (line 49, first paragraph)? How different is it from the national survey?

Response 9: We have included information about the age cohort covered in the MICS survey and clarified the differences between the MICS survey and the national survey. Thank you for bringing up this point. (Line 52 in the manuscript with track changes version).

Comment 10: Page 3 lines 52-53, there is not much difference between rural and urban areas based on the figures reported (2.9% vs 2.5 %)? How precise are those estimates, what are the confidence intervals and/or p-value?

Response 10: We appreciate your comment. Unfortunately, the report from which these data were extracted did not provide precision measures such as confidence intervals or p-values. Hence, we were unable to present them in the manuscript.

Comment 11: Lines 54-62- The sentences are very long with several grammatical errors. The whole paragraph should be revised.

Response 11: We have thoroughly revised the entire paragraph as per your recommendation, addressing the issues of sentence length and grammatical errors. Thanks. (Line 57-74 in the manuscript with track changes version).

Comment 12: Based on the information provided in lines 63-78, the association between the COVID 19 pandemic and mental health is well established. What does your study add? How different is it to other studies? What is the significance of your study in the context of Bangladesh?

Response 12: Thank you for your comments. Throughout the discussion, we have explored the main findings of our study and compared them with other studies. As this is the first study of its kind in Bangladesh, we were unable to directly compare it with other Bangladeshi studies. However, we have further emphasized the significance of our study in the context of Bangladesh, addressing the points you raised. Please refer to the last paragraph of the discussion for more information. (Line 341-352 in the manuscript with track changes version).

Comment 13: Lines 180-199, what is reported in Table 2 and Figure 1 are not prevalence rates, please revise the text accordingly.

Response 13: We have made the necessary revisions in the text to accurately describe the content of Table 2 and Figure 1 in relation to prevalence. Thank you for bringing this to our attention. (Line 202, 203, 211 in the manuscript with track changes version).

Comment 14: Line 263-281, the discussions on marital status are contradictory.

Response 14: We apologize for any confusion caused. Since the prevalence of mental health issues among singles was not statistically significant in our regression model. Hence, we have removed the contradictory discussions on marital status from the manuscript. (Line 296-299 in the manuscript with track changes version).

Comment 15: There are some grammatical, typo and formatting errors.

Response 15: We have carefully addressed the grammatical, typo, and formatting errors in the revised manuscript. Thank you for bringing them to our attention.

---

## [Editor Report · Decision Letter 1]

19 Jun 2023

PONE-D-23-05590R1Prevalence and factors associated with depression, anxiety, and stress among people with disabilities during COVID-19 pandemic in Bangladesh: a cross-sectional studyPLOS ONE

Dear Dr. Roy,

Thank you so much for submitting the revised version. It looks very good now. I have thoroughly read the manuscript. I would request to have an editorial service to make the manuscript more coherent and improved. The minimum you can do probably is to use online editorial services such as grammarly to improve the language. I think it is in good shape; however, I hope you would agree that before going to publication, you can probably take the opportunity to improve it as much as possible. Please resubmit the manuscript after those revisions

We look forward to receiving your revised manuscript.

Kind regards,

Shafiun Nahin Shimul

Academic Editor

PLOS ONE
---

## [Author Response · Author response to Decision Letter 1]

22 Jun 2023

Shafiun Nahin Shimul

Academic Editor

PLOS ONE

Dear Editor,

Thank you very much for reviewing our manuscript titled: “Prevalence and factors associated with depression, anxiety, and stress among people with disabilities during COVID-19 pandemic in Bangladesh: a cross-sectional study”. 

Thank you so much for giving us the chance to improve the quality of our manuscript. We further modified our manuscript substantially to make it more coherent and enhanced (Please check the with track changes version of our manuscript). In the discussion, we also added one more reference.

We hope that our manuscript will be acceptable for publication in “PLOS ONE”.

Yours Sincerely,

Nitai Roy, PhD

Corresponding author

---

## [Editor Report · Decision Letter 2]

26 Jun 2023

Prevalence and factors associated with depression, anxiety, and stress among people with disabilities during COVID-19 pandemic in Bangladesh: a cross-sectional study

PONE-D-23-05590R2

Dear Dr. Roy,

We’re pleased to inform you that your manuscript has been judged scientifically suitable for publication and will be formally accepted for publication once it meets all outstanding technical requirements.

Kind regards,

Shafiun Nahin Shimul

Academic Editor

PLOS ONE
---

## [Editor Report · Acceptance letter]

28 Jun 2023

PONE-D-23-05590R2 

Prevalence and factors associated with depression, anxiety, and stress among people with disabilities during COVID-19 pandemic in Bangladesh: a cross-sectional study 

Dear Dr. Roy:

I'm pleased to inform you that your manuscript has been deemed suitable for publication in PLOS ONE. Congratulations! Your manuscript is now with our production department. 

Kind regards, 

on behalf of

Dr. Shafiun Nahin Shimul 

Academic Editor

PLOS ONE